# Gastroesophageal Reflux Disease Symptoms after Laparoscopic Sleeve Gastrectomy: A Retrospective Study

**DOI:** 10.3390/jpm12111795

**Published:** 2022-10-31

**Authors:** Wen-Yang Wu, Shih-Chun Chang, Jun-Te Hsu, Ta-Sen Yeh, Keng-Hao Liu

**Affiliations:** 1Division of General Surgery, Department of Surgery, Linkou Chang-Gung Memorial Hospital, Taoyuan 333, Taiwan; 2School of Medicine, College of Medicine, Chang Gung University, Taoyuan 333, Taiwan

**Keywords:** laparoscopic sleeve gastrectomy, gastroesophageal reflux disease, morbid obesity, bariatric surgery

## Abstract

(1) Background: Laparoscopic sleeve gastrectomy (LSG) is widely performed in bariatric surgery. However, the prevalence and risk factors of gastroesophageal reflux disease (GERD) symptoms after LSG remain unclear to date. This study aimed to identify risk factors of GERD after LSG. (2) Methods: We conducted a retrospective study at Linkou Chang Gung Memorial Hospital and reviewed 296 patients who underwent LSG from 2016 to 2019. A total of 143 patients who underwent preoperative esophagogastroduodenoscopy and completed the 12-month postoperative follow-up were enrolled. Patients’ demographic data, comorbidities, and postoperative weight loss results were recorded for analysis. The GerdQ questionnaire was used to assess GERD after LSG. (3) Results: There were eight surgical complications (5.6%) among the 143 studied patients (median age, 36 years; 56 (39.2%) men; median body weight 105.5 kg; median body mass index [BMI], 38.5 kg/m^2^). Twenty-three patients (16.1%) developed de novo GERD symptoms. GERD was significantly associated with older age (*p* = 0.022) and lower BMI (<35 kg/m^2^, *p* = 0.028). In multiple logistic regression analysis, age and BMI were significantly associated with GERD. (4) Conclusions: LSG is a safe and effective weight loss surgery. In our study, it led to 16.1% of de novo GERD symptoms, which were significantly related to older age and lower BMI (<35 kg/m^2^).

## 1. Introduction

Obesity is a modern-day global epidemic that is causing economic burden and a significant impact on health. In 2016, 39% of adults aged 18 years and older (39% of men and 40% of women) were overweight. Approximately 13% of the adult population (11% of men and 15% of women) were obese [1,2]. In Taiwan, a recent study showed that 47.97% of adults were overweight in 2020 [3].

Bariatric surgery is the most effective means of obtaining substantial and lasting weight loss in individuals with obesity [4]. Among the bariatric surgery options, laparoscopic sleeve gastrectomy (LSG) became the most frequently performed bariatric procedure to date because of its effectiveness in weight loss, relative simplicity, and fewer long-term nutritional problems than other bariatric procedures [5]. Sleeve gastrectomy was first introduced in 1988 by Hess as part of the biliopancreatic diversion duodenal switch procedure. In 1999, it was first performed laparoscopically and gradually became a stand-alone bariatric procedure [6].

Although LSG provides good weight loss results, there are controversial effects on de novo gastroesophageal disease (GERD) [7]. Previous studies reported that the incidence of new-onset GERD ranges from 0 % to 34.9% in Western countries [8], and the risk factors of de novo GERD after LSG are still controversial. Additionally, the incidence and risk factors of de novo GERD after LSG in current literature from the Asia-Pacific region remain unclear to date. Therefore, this study aimed to identify the incidence and notable risk factors of GERD after LSG.

## 2. Materials and Methods

### 2.1. Ethics Statements

The protocol for this study was reviewed and approved by Chung Gung Medical Foundation Institute Review Board (IRB No.: 202200918B0).

### 2.2. Study Design and Population

The study was a retrospective analysis of prospectively collected data on morbidly obese patients from Linkou Chang Gung Memorial Hospital (CGMH). Surgical criteria for morbid obesity were based on the 2005 Asia-Pacific Bariatric Surgery Group consensus meeting [9]. Asian patients with a body mass index (BMI) more than 37.5 kg/m^2^ or more than 32.5 kg/m^2^ with obesity-related comorbidities, such as type II diabetes (glycated hemoglobin A1C level ≥7.5%), hypertension, obstructive sleep apnea, etc., were candidates for bariatric surgery. Patients who underwent esophagogastroduodenoscopy (EGD) preoperatively were selected for inclusion in our study. Patients’ medical histories were carefully reviewed, and those who took medication for heartburn and/or regurgitation; or those with evidence of GERD before surgery (grade C or D according to the Los Angeles [LA] classification) due to preoperative EGD, were excluded.

### 2.3. Procedures

EGD was not a routinely recommended procedure in the workup for bariatric surgery. The American Society of Metabolic and Bariatric Surgery advised the use of endoscopy only for patients with significant gastrointestinal symptoms [10,11]. In our hospital, some surgeons preferred performing EGD for patients before surgery. LSG was the first considered bariatric procedure in our department for all morbidly obese patients. However, laparoscopic Roux-en Y gastric bypass (LRYGB) was recommended for patients with type II diabetes older than 8 years of age and those who needed insulin injections or a preoperative C peptide level less than 2 mmol/L. Furthermore, LRYGB was also recommended for patients with preoperative GERD symptoms, those with the need for prolonged proton pump inhibitor (PPI) treatment, those with evidence of GERD (grade C or D according to the Los Angeles [LA] classification), and those with a hiatal hernia due to preoperative EGD. Postoperatively, we measured the symptoms and signs and scheduled EGD at 12 months postoperatively if the patient had severe GERD symptoms.

### 2.4. Definitions

The weight loss result was calculated based on the percentage of excess weight loss (%EWL) and percentage of total weight loss (%TWL). Patients who reach %EWL ≥ 50 or %TWL ≥ 20% at postoperative month 12 were defined as achieving successful weight loss [12,13].

### 2.5. GerdQ Questionnaire

Most of the patients did not want to undergo invasive examination such as EGD or functional studies (pH-manometry) after surgery. We used more convenient tools to evaluate GERD, considering the cost effectiveness and time efficiency. GERD symptoms were assessed by administering the GerdQ questionnaire during the postoperative outpatient follow-up. The GerdQ questionnaire is a simple, self-administered and patient-centered questionnaire that includes six items [14]. It asks patients to score the number of days with symptoms and use of over-the counter (OTC) medications during the previous 7 days. It uses a four-point Likert scale (0–3) to score the frequency of four positive predictors of GERD (heartburn, regurgitation, sleep disturbance due to reflux symptoms, or use of OTC medications for reflux symptoms) and a reversed Likert scale (3–0) for two negative predictors of GERD (epigastric pain and nausea); the total GerdQ score ranges from 0 to 18 [14]. Patients completed the GerdQ questionnaire at 12 months after LSG. De novo GERD is defined as a GerdQ score ≥9 as per literature [14].

### 2.6. Laparoscopic Sleeve Gastrectomy

LSG was performed using the same technique between January 2016 and December 2019. A 32-French bougie was applied for calibration of the gastric tube. Dissection was started 4 cm proximal to the pylorus and proceeded to 1 cm from the angle of His. We did not check the hiatus routinely since the presence of a hiatal hernia was excluded by preoperative EGD. All patients received seromuscular suturing for staple line reinforcement, and no leak test was performed routinely intra- and postoperatively. Patients were instructed to follow a clear liquid diet on the first postoperative day, followed by a pureed diet on the second day. In general, patients were discharged on postoperative day 3, if they had stable vital signs and adequate pain control.

### 2.7. Statistical Analysis

Data are presented as median (interquartile range) or percentages. Patients’ characteristics were compared using the Mann–Whitney U test for nonparametric data and the chi-square test for categorical data. Logistic regression analysis was performed to calculate odds ratios (ORs) for predictive factors. When the data reached statistical significance in invariant analysis, multivariate analysis with logistic regression was conducted to determine the relationship between predictive factors and de novo GERD symptoms. Data were analyzed using SPSS Statistics, version 26 (IBM Corp.). Variables with a 95% CI for the OR that did not include 1.0 and *p*-values < 0.05 were considered to be significant.

## 3. Results

### 3.1. Patient Characteristics

In total, 384 patients who received bariatric surgery at Linkou CGMH between January 2016 and December 2019 were screened. Among those patients, 143 patients were eligible for the study. The study flow chart is shown in Figure 1.

Patients’ clinical characteristics are summarized in Table 1. The median age at surgery was 36 years. Most participants were female (60.8%). The baseline body weight and BMI were 105.5 kg and 38.5 kg/m^2^, respectively. Twenty-seven (18.9 %) patients reported tobacco use, and 15 (10.5 %) reported alcohol consumption. The most common comorbidity was fatty liver (87.4%), followed by hypertension (46.2%), diabetes mellitus (20.3%), and dyslipidemia (36.4%). Preoperative EGD revealed LA grade A esophagitis in 37.1% of patients, LA grade B esophagitis in 2.8%, gastric ulcer in 39.9%, gastritis in 85.3%, small hiatal hernia in 2.8%, and Helicobacter pylori infection in 21.7%. Eight (5.6%) patients had complications, such as leakage (2, 1.4%), surgical site infection (1, 0.7%), postoperative bleeding (1, 0.7%), ventral hernia (3, 2.1%), and ileus (1, 0.7%) after LSG. Reoperation to assess for bleeding and enterolysis was performed accordingly. There was a significant reduction in the BMI before (38.5 kg/m^2^) and after (27.1 kg/m^2^) LSG with %EWL and %TWL of 68.0 ± 22.9% and 30.6 ± 10.4%, respectively.

### 3.2. Evaluation of Gastroesophageal Disease Postoperatively

PPI use was recorded postoperatively through the 12-month follow-up. Overall, 141 patients (98.6%) in postoperative month 1 and 46 (32.2%), 35 (24.8%), 30 (21.0%), and 29 (20.3%) patients in postoperative months 3, 6, 9, and 12, respectively, could not stop using PPIs.

Regarding the results of the GerdQ questionnaire in postoperative month 12, 23 patients (16.1%) were diagnosed with de novo GERD symptoms after LSG. The risk factors for predicting de novo GERD symptoms are shown in Table 2. In invariant analysis, de novo GERD symptoms were not associated with tobacco use and alcohol consumption. However, de novo GRED symptoms were associated with older age (40 years versus [vs.] 35 years, *p* = 0.005) and lower BMI (<35 kg/m^2^ vs. ≥35 kg/m^2^, *p* = 0.019). Preoperative comorbidities, such as hypertension, diabetes mellitus, and fatty liver, and the postoperative weight loss result were not associated with de novo GERD symptoms.

Age and preoperative BMI were included in multivariable analysis with logistic regression. The results are shown in Table 3. De novo GERD symptoms were significantly associated with older age (OR = 1.1; *p* = 0.022) and lower BMI (OR = 3.1; *p* = 0.028).

## 4. Discussion

The main findings of our study are as follows. (1) After LSG, the incidence of de novo GERD symptoms was 16.1%. (2) Older patients and those with a lower preoperative BMI have significant risk of de novo GERD postoperatively. LSG is effective in terms of weight loss, relative simplicity, and fewer long-term nutritional complications than other bariatric surgeries [6]. Therefore, LSG became the current leading bariatric procedure [6].

Various modalities for evaluating GERD after LSG were reported, and the incidence of GERD after LSG is up to 34.9% [8]. In 2012, Chopra et al. reported a postoperative GERD rate of 3.7% by chart review [15]. In a postoperative EGD study, Soricelli et al. reported an incidence of 15.9% in 2013 [16]. In another study, the barium swallow, GerdQ questionnaire, and 24 h pH-manometry were applied, and the incidence was approximately 11% [17,18,19]. In our study, we evaluated de novo GERD by the GerdQ questionnaire and found an incidence of 16.1%. The GerdQ questionnaire is a simple, non-invasive, self-administered, patient-centered tool for the diagnosis of GERD. The implementation of GerdQ could reduce the need for upper endoscopy and improve resource utilization [14].

The putative pathophysiological mechanisms of de novo GERD after LSG include hypotensive lower esophageal sphincter (LES), an increased gastroesophageal pressure gradient, intrathoracic sleeve migration, and relative gastric stasis in the proximal remnant [2,20,21]. De novo hiatal hernia of the gastric tube after sleeve gastrectomy, which is also named intrathoracic sleeve migration, confirmed causes of GERD [22,23]. We did not routinely perform EGD or barium swallowing for the diagnosis of de novo GERD. However, in some patients with persistent and severe symptoms of de novo GERD, EGD and barium swallowing showed a newly developed hiatal hernia, which was not noted preoperatively.

Similar to the treatment of GERD patients without bariatric procedure, the initial management of those with de novo GERD include dietary and lifestyle modification, alcohol and smoking cessation, and PPI prescription, although most symptoms of de novo GERD symptoms cannot be controlled by PPIs. In our study, 87 patients needed PPIs in postoperative month 3, and 33 patients stopped taking them in postoperative month 12. Fourteen patients had GERD symptoms and five patients were well controlled with PPIs over the next few months.

For patients with severe symptoms who failed conservative management, several endoscopic procedures, such as plication, bulking the LES with inert biopolymers, and thermal ablation, were used for treating de novo GERD [24]. These procedures targeted the muscular layer of the LES through endoscopic means to decrease esophageal sensitivity to acid and reduce gastroesophageal junction compliance [24,25]. When it comes to surgical options, revision surgery can correct anatomic abnormalities (i.e., hiatal hernia, intragastric migration of the sleeve, distal stenosis or torsion angulation of the sleeve, and retained fundus) [26,27]. LRYGB is the most effective surgical treatment for GERD in patients with obesity [25]. LRYGB can almost completely resolve GERD and de novo GERD symptoms [25,28].

The aforementioned treatment should be openly discussed with the patients when making decisions about surgery, medications, or self-management. The opportunity to prevent such patients from having persistent gastroesophageal reflux should be seized, and the available evidence of treatment should be openly discussed with the patients.

The main finding of our study was that preoperative BMI and age were risk factors for de novo GERD after LSG. Population-based studies also found an association between GERD and higher BMI. Similar to our study, Althuwaini et al. reported that 213 patients who underwent LSG and had a high preoperative BMI were less likely to develop new-onset or worsening symptoms of GERD [29]. However, two other studies showed no significant association between the preoperative BMI and development of de novo GERD after LSG [30,31]. The association with preoperative BMI has to be interpreted with caution, since it was not the primary endpoint of our study. Nonetheless, this finding should be explored in future studies.

One study that included 326 patients reported the same result for older age and the occurrence of postoperative GERD symptoms that our study showed [32]. Age was associated with an increase in esophageal acid exposure. Aging is attributed with a progressively decreased abdominal LES length and esophageal motility. The severity of GERD increased in the elderly population due to degradation of the gastroesophageal junction and impaired esophageal clearance [33]. The physiological change supports ours finding and addresses the effect of age on developing de novo GERD after SLG.

### Limitations

This study has some limitations. First, we used the GerdQ questionnaire to evaluate postoperative GERD symptoms. It is not objective and is not the gold standard test for diagnosing GERD. Patients with silent reflux or delayed onset of reflux were potentially missed. Although EGD or 24 h pH-manometry is the gold standard examination, considering its cost effect and time efficiency, we did routinely perform such examinations, except in patients with severe GERD symptoms. Second, inconsistent assessment before and after operation made evidentiary uncertainty. In the future, we will design a more rigorous method to avoid different inspections for comparison. Third, our study population was from a single institution; thus, it cannot represent the whole Asian-Pacific population, and this factor may cause bias in the results. Third, our study was a retrospective study and only had a 1-year follow-up period. Further prospective studies and a long-term follow-up period are needed in the future to obtain a more convincing conclusion.

## 5. Conclusions

In conclusion, LSG is a safe and effective weight loss surgery. However, further prospective studies are needed to determine the risk factors associated with post-LSG GERD and overall quality of life.

## Figures and Tables

**Figure 1 jpm-12-01795-f001:**
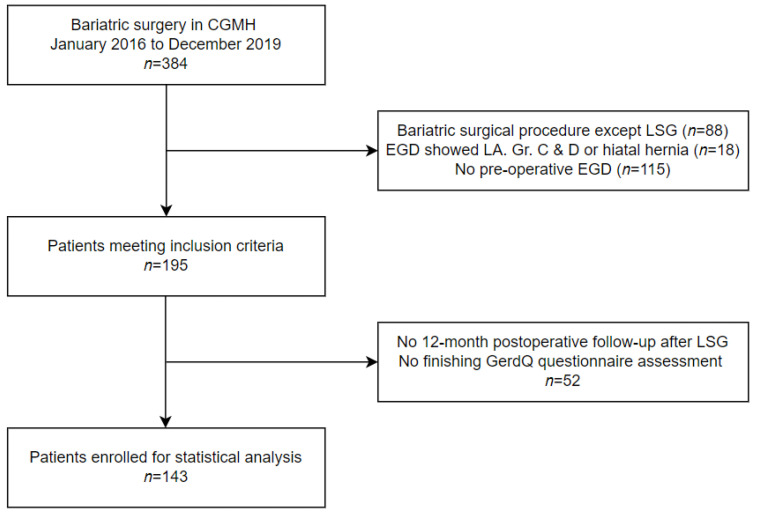
Study flow chart. CGMH, Chang Gung Memorial Hospital; LSG, laparoscopic sleeve gastrectomy; EGD, esophagogastroduodenoscopy; LA Gr., Los Angeles grade.

**Table 1 jpm-12-01795-t001:** Patients’ baseline characteristics.

Characteristic	No. of Patients	Percentage (%) or Median (IQR)
Patients	143	
Sex		
Female	87	60.8
Male	56	39.2
Age (years)		36 (12.5)
Tobacco use	27	18.9
Alcohol consumption	15	10.5
** *Body composition* **		
Weight (kg)		105.5 (27)
Height (cm)		165.0 (14)
BMI (kg/m^2^)		38.5 (7.4)
BMI category (kg/m^2^)		
<35	28	19.6
≥35	115	80.4
** *Comorbidity* **		
Hypertension	66	46.2
Diabetes mellitus	29	20.3
Dyslipidemia	52	36.4
Fatty liver	125	87.4
Gout	11	7.7
** *EGD before LSG* **		
LA grade for GERD		
A	53	37.1
B	4	2.8
Gastric ulcer	57	39.9
Gastritis	122	85.3
Hiatal hernia	4	2.8
HP infection	31	21.7
** *Result after LSG* **		
Complication	8	5.6
Weight (kg)		73.0 (17.9)
BMI (kg/m^2^)		27.1 (5.6)
%EWL at 12 months postoperatively		
Average		68.0 (22.9)
<50%	20	14.0
≥50%	123	86.0
%TWL at 12 months postoperatively		
Average		30.6 (10.4)
<20%	14	9.8
≥20%	129	90.2
PPI		
None	95	66.4
PRN	19	13.3
Continuous	29	20.3
Presence of GERD (GerdQ score)		
<9	120	83.9
≥9	23	16.1

Abbreviations: no., number; IQR, interquartile range; BMI, body mass index; EGD, esophagogastroduodenoscopy; LSG, laparoscopic sleeve gastrectomy; GERD, gastroesophageal disease; LA. Grade, Los Angeles grade; HP, *Helicobacter pylori*; %EWL, percentage of excess weight loss; %TWL, percentage of total weight loss; PPI, proton pump inhibitor use within 12 months postoperatively; PRN, pro re nata.

**Table 2 jpm-12-01795-t002:** Univariate analysis of the predictive factors of de novo GERD after LSG.

Characteristic		GERD (12 Months Postoperatively)	Non-GERD (12 Months Postoperatively)	*p*-Value
** *No. of Patients* **		23	120	
** *Basic Data* **				
Age (years)	Median (IQR)	40 (12)	35 (13)	0.005
Sex	Male	10 (17.9)	46 (82.1)	0.643
	Female	13 (14.9)	74 (85.1)	
BMI (kg/m^2^)	<35	9 (32.1)	19 (67.9)	0.019
	≥35	14 (12.2)	101 (87.8)	
Tobacco use	Yes	6 (22.2)	21 (77.8)	0.383
	No	17 (14.7)	99 (85.3)	
Alcohol consumption	Yes	4 (26.7)	11 (73.3)	0.264
	No	19 (14.8)	109 (85.2)	
** *Comorbidity* **				
Hypertension	Yes	13 (19.7)	53 (80.3)	0.276
	No	10 (13.0)	67 (87.0)	
Diabetes mellitus	Yes	3 (10.3)	26 (89.7)	0.412
	No	20 (17.5)	94 (82.5)	
Dyslipidemia	Yes	9 (17.3)	43 (82.7)	0.763
	No	14 (15.4)	77 (84.6)	
Fatty liver	Yes	21 (16.8)	104 (83.2)	0.738
	No	2 (11.1)	16 (88.9)	
Gout	Yes	3 (27.3)	8 (72.7)	0.384
	No	20 (15.2)	112 (84.8)	
** *EGD before LSG* **				
LA grade of GERD	None	14 (16.3)	72 (83.7)	0.871
	Grade A	8 (15.1)	45 (84.9)	
	Grade B	1 (25.0)	3 (75.0)	
Gastric ulcer	Yes	10 (17.5)	47 (82.5)	0.699
	No	13 (15.1)	73 (84.9)	
Gastritis	Yes	19 (15.6)	103 (84.4)	0.748
	No	4 (19.0)	17 (81.0)	
Hiatal hernia	Yes	2 (50.0)	2 (50.0)	0.121
	No	21 (15.1)	118 (84.0)	
HP infection	Yes	5 (16.1)	26 (83.9)	1.000
	No	18 (16.1)	94 (83.9)	
** *Result after LSG* **				
Complication	Yes	1 (12.5)	7 (87.5)	>0.999
	No	22 (95.7)	113 (94.2)	
EWL (%)	<50	4 (20.0)	16 (80.0)	0.531
	≥50	19 (15.4)	104 (84.6)	
TWL (%)	<20	3 (21.4)	11 (78.6)	0.700
	≥20	20 (15.5)	109 (84.5)	

Data are expressed as numbers with percentages in parentheses, unless otherwise stated. A *p*-value < 0.05 was considered statistically significant. Abbreviations: no., number; GERD, gastroesophageal disease; IQR, interquartile range; BMI, body mass index; EGD, esophagogastroduodenoscopy; LSG, laparoscopic sleeve gastrectomy; LA grade, Los Angeles grade; HP, *Helicobacter pylori*; EWL, excess weight loss; TWL, total weight loss.

**Table 3 jpm-12-01795-t003:** Results of multivariable analysis with logistic regression.

Predictor Variable	OR	95% CI		*p*-Value
		Lower	Upper	
Age (years)	1.057	1.008	1.108	0.022
BMI (kg/m^2^)				
<35	3.067	1.130	8.327	0.028
≥35	Reference			

A *p*-value < 0.05 was considered statistically significant. Abbreviations: BMI, body mass index; and OR, odds ratio.

## Data Availability

The raw data supporting the conclusions of this article will be made available by the authors, without undue reservation.

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
