# Peer review of "Gastroesophageal Reflux Disease Symptoms after Laparoscopic Sleeve Gastrectomy: A Retrospective Study"

_jpm, 2022, doi:10.3390/jpm12111795_

Round 1

Reviewer 1 Report

Thank you for letting me review your article

I want to congratulate the author on submitting this paper on the important topic of the relationship between laparoscopic sleeve gastrectomy and de novo GERD

The study evaluate the incidence of de novo GERD post sleeve gastretomy

My main concern is regarding the methodology of the study.

I didn't understand what was the pre operative assessment regarding the presence of GERD. Did you ask specific questions ? Did you review recent medication ?

The fact that you didn't performed GerdQ questionnaire before hand weakened this study. With that being said, If you'll be able to elaborate on this topic, It will strengthen the study.

The second subject is the relationship of de novo GERD and age. In your univariate analysis you wrote that older age was related to de novo GERD (line 144) . Your multivariate analysis found OR of 1.057 which is minimal. Have you tried creating two age groups in to find better results ? 

Reviewer 2 Report

This is a well written manuscript regarding a very important topic in Bariatric/Foregut Surgery and I commend the authors for exploring there topic. 

I do have some concerns:

The overall findings of study represent SUBJECTIVE improvement and this should be clearly stated.

Why did some patients undergo a preoperative EGD and some not? Is there any reason for performing it or not performing it? The reason/indication for acquiring/not acquiring a pre-operative EGD should be stated. It should also be added to the limitations section as a selection bias.

Was an upper GI fluoroscopy study acquired on any patients? If so, what were the findings.

How was pre-operative GERD assessed? many patients with morbid obesity suffer from GERD pre-operatively and assuming post-operative De-Novo GERD requires knowing how many patients suffered from GERD pre-operatively. It appears that in your study none of them had pre-operative GERD, which means either it is under-powered, or there is a selection bias. If patients with pre-op GERD were excluded from the study , it should be clearly stated, not just mention their referral for a RYGB. What about all the patients that didn't undergo an EGD? how was their operation chosen?

Was there any Barrette's esophagus found on any EGD(pre- or post-op)?

Why was a score of 9 selected as a cut-off for diagnosis of GERD? If a subjective test was used, why was it not used pre-operatively as well? It could show improvement/worsening of symptoms according to the same scale. Aside from the lack of objective studies (pH manometric studies), this is significant drawback of the study.

Did you only examine patients with GERD symptoms post-operatively? Patients with silent reflux or delayed onset of reflux are missed and this should be stated in the limitations of the study. 

minor revisions:

Percentages of complications should be calculated out of the total operated population(n=143). For instance, leakage rate was 1.4% (which is higher than acceptable) and not 25%.

Line 160: "fewer long‐term nutritional complications than other bari‐ 160 atric surgeries."

Your study does not evaluate this.

Reviewer 3 Report

This is a paper regarding GERD symptoms after LSG and not de-novo GERD and this should be mentioned in title

GERD diagnosis is usually based on the combination of symptoms, endoscopic findings and pH manometry, because it is well know in literature tha nearly 50% of pts che have symptoms without any other findings (see paper from Patti et al and so on).

The major flaw is that Authors did administered neither GERDQ questionnaire pre and post-operatively nor post-op EGD to match the GERD questionnaire with symptoms and findings. No mention at all to functional studies (pH-manometry)

Line 58: this consensu Meeting should be cited as published reference

Round 2

Reviewer 1 Report

The main issue in this paper is the fact that the preoperative assessment for GERD and the postoperative assessment for GERD are not the same

I understand the difficulties in performing an EGD beforehand. With that being said, a GerdQ questionnaire before surgery would have help this comparison

Reviewer 2 Report

Comments have been addressed appropriately.

Reviewer 3 Report

limitations of this study have been pointed out. Nothing to add.
